# Experimental Investigation of a Bipartite Quench
# in a 1D Bose gas

L. Dubois[1], G. Themèze[1], J. Dubail[2] and I. Bouchoule[1],

**1** Laboratoire Charles Fabry, Institut d'Optique, CNRS, Université Paris-Saclay
**2** CESQ and ISIS (UMR 7006), University of Strasbourg and CNRS, 67000 Strasbourg, France
\* lea.dubois@universite-paris-saclay.fr

May 31, 2025

## Abstract

Long wavelength dynamics of 1D Bose gases with repulsive contact interactions can be captured by Generalized HydroDynamics (GHD) which predicts the evolution of the local rapidity distribution. The latter corresponds to the momentum distribution of quasiparticles, which have infinite lifetime owing to the integrability of the system. Here we experimentally investigate the dynamics for an initial situation that is the junction of two semi-infinite systems in different stationary states, a protocol referred to as 'bipartite quench' protocol. More precisely we realise the particular case where one half of the system is the vacuum state. We show that the evolution of the boundary density profile exhibits ballistic dynamics obeying the Euler hydrodynamic scaling. The boundary profiles are similar to the ones predicted with zero-temperature GHD in the quasi-BEC regime, with deviations due to non-zero entropy effects. We show that this protocol, provided the boundary profile is measured with infinite precision, permits to reconstruct the rapidity distribution of the initial state. For our data, we extract the initial rapidity distribution by fitting the boundary profile and we use a 3-parameter ansatz that goes beyond the thermal assumption. Finally, we investigate the local rapidity distribution inside the boundary profile, which, according to GHD, presents, on one side, features of zero-entropy states. The measured distribution shows the asymmetry predicted by GHD, although unelucidated deviations remain.

# 1 Introduction

Gaining insight on the out-of-equilibrium dynamics of many-body quantum systems is tremendously difficult and is the goal of an active research field. One particular class of systems where important progress has been made over the past decade is the class of integrable one-dimensional systems. Owing to their infinite number of local conserved charges, the description of the local properties of stationary states that arise after relaxation requires a whole function, the rapidity distribution [1–7]. This function can be viewed as the velocity distribution of the infinite-lifetime quasi-particles in the system. Its large-scale effective dynamics is described by generalized hydrodynamics (GHD) [8, 9] (for recent reviews, see e.g. [10, 11]), which assumes local relaxation to a local stationary state. As with any hydrodynamic theory, the most paradigmatic situation that can be handled by GHD is the 'Riemann problem' [12], also dubbed 'bipartite quench' more recently [13–18], or 'domain-wall quench' or 'domain-wall protocol' [19–27]. In this 'bipartite quench protocol', the microscopic dynamics is governed by a translation-invariant Hamiltonian but the initial state is the junction of two semi-infinite homogeneous systems each prepared in a different stationary state of the Hamiltonian. The GHD theory predicts that, at times long enough such that diffusion effects become negligible [28] and Euler-scale hydrodynamics is valid, the time evolution is ballistic. An interesting feature of this protocol is that the local state, within the merging region, is expected to present features characteristic of zero-temperature systems. Thus, this protocol could be used to reveal power-law singularities of correlation functions characteristic of a zero-temperature Luttinger liquid [13], provided a local probe is used.

In this paper, we experimentally realize an instance of the bipartite quench protocol using an ultra-cold atomic Bose gas, well described by the Lieb-Liniger model of one-dimensional Bosons with contact repulsive interactions [29, 30], which is an integrable model. In our experiment, the bipartition consists of the junction of a gas in a stationary state on one side, and the vacuum on the other side. This initial state is prepared by producing a homogeneous atomic cloud and by removing suddenly its left part. For different evolution times, we record the density profile of the boundary between the two regions, dubbed the boundary profile. We find that the boundary profile exhibits a ballistic behavior, as expected from the predictions of GHD theory at the Euler scale.

The boundary profile, for clouds prepared with deep evaporative cooling, is in fair agreement with GHD predictions assuming the semi-infinite gas is in its ground state, although deviations are present. From the boundary profile, we show that it is in principle possible to reconstruct the rapidity distribution characterizing the initial gas. This protocol can thus be used as a generalized thermometry. However, the reconstruction method suffers from a high sensitivity to experimental noise in the tail of the boundary profile, which prevents us from reconstructing faithfully the initial rapidity distribution. Instead, we use an ansatz parametrized by a few parameters to extract the rapidity distributions of the initial gas from a fit to the boundary profile.

Finally, we use a newly developed technique [31] to probe the local rapidity distribution within the boundary. The latter is expected to be highly asymmetric for an initial state whose rapidity distribution is substantially broader and smoother than that of the ground state: while one of its borders reflects the broad character of the initial rapidity distribution, the other border presents the sharp feature expected for the ground state. Our experimental data show

such an asymmetric behavior, although the sharp border presents an unelucidated tail.

## 2   Experimental setup

We produce an ultra-cold gas of $^{87}$Rb bosonic atoms in the stretched state $|F = 2, m_F = 2\rangle$ using an atom chip. In addition to a homogeneous longitudinal magnetic field $B_0 = 3.36$G, transverse trapping is achieved with three parallel microwires deposited on the chip (shown in blue in Fig.1(a)) which carry AC currents modulated at 400MHz. This configuration eliminates wire roughness effects and allows independent control over both longitudinal and transverse confinement [32]. The atoms are trapped $7\mu$m from the chip surface and $15\mu$m from the wires, enabling strong transverse confinement. The transverse trapping potential is well approximated by a harmonic potential with a frequency of $\omega_\perp/2\pi = 2.56$kHz for the data presented in this paper. Using radio-frequency evaporative cooling, we produce an atomic cloud at a temperature of approximately $T = 100$nK and a chemical potential of $\mu/k_B = 45$nK. With these parameters, $\mu/(\hbar\omega_\perp) = 0.4$ and $k_B T/(\hbar\omega_\perp) = 0.8$, the gas enters in the 1D regime. The effective 1D coupling constant for atoms in the transverse ground state is given by $g = 2a_{3D}\hbar\omega_\perp$ [33], where $a_{3D} = 5.3$ nm is the 3D scattering length of $^{87}$Rb [34]. Further details on the setup can be found in [35]. The dimensionless Lieb parameter $\gamma = mg/(\hbar^2 n_0)$, where $n_0$ is the linear density, lies in the range $[0.4, 0.7] \times 10^{-2}$ and the temperature fulfills $T \ll n_0^{3/2}\sqrt{\hbar^2 g/m}/k_B$ such that the atomic clouds produced are deeply in the quasicondensate regime [36].

The longitudinal magnetic trap is produced by DC currents running through four wires positioned on either side of the three microwires, as shown in the Fig.1(a). Since these wires are placed far from the atomic cloud, the longitudinal potential can be expressed as a polynomial series expansion $V(x) = \sum_i a_i x^i$. The fourth first coefficients $a_i$ are tuned by adjusting the currents in the four wires that generate the longitudinal trapping potential. By carefully selecting these currents, it is possible to set $a_1$, $a_2$ and $a_3$ to zero such that the leading term is the quartic term $V(x) = a_4 x^4$. Such a potential permits to achieve a quasi-homogeneous atomic density over a relatively large region, an important feature to study the bipartite quench protocol which assumes a semi-infinite system. An example of linear density profile for an atomic cloud placed in such a potential is represented in gray in Fig.1(b). The linear density $n_0$ remains constant to within 10% around the peak density over a range of approximately $250\mu$m.

To produce the initial bipartition, we use the selection method introduced in [37]. We illuminate the left border of the atomic cloud, initially in a global stationary state in a quartic trap, with a pushing beam that is nearly resonant with the $F = 2 \rightarrow F' = 3$ transition of the $D2$ line and which propagates perpendicularly to $x$. Atoms shined by this pushing beam are subjected to radiation pressure : after being illuminated for $30\mu s$ corresponding to $\sim 15$ absorption/reemission cycles, atoms have enough energy to leave the trap. To illuminate only a border of the gas, the beam is shaped using a digital micromirror device (DMD). Further details on this spatial selection method are available in [37]. This protocol produces a sharp boundary between a zero density system and a quasi-homogeneous gas due to the fact that the atoms are initially placed in a quartic trap. The sharpness of the boundary is mainly limited by the imaging resolution, which is in the micrometer range. The reabsorption of scattered photons by the atoms which are not shined could also limited the boundary sharpness. This effect is mitigated by detuning the pushing beam by 15MHz from the D2 transition. An example of the density profile of a gas initially in a global stationary state in a quartic trap, after applying this spatial selection tool, is shown in yellow in Fig.1(b).

The longitudinal confinement is then removed while maintaining the transverse confinement. The initial sharp boundary broadens in time and this dynamics is monitored by recording longitudinal density profiles $n(x, t)$ after different evolution time $t$.

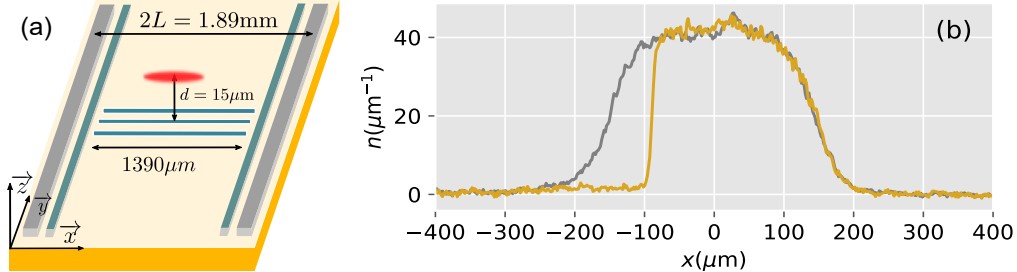

Figure 1: (a) Schematic drawing of the atom chip. The 3 blue wires produce the transverse trapping, the 4 other wires produce the longitudinal trapping. The red oval ball represents the atomic cloud, trapped 12 microns above the wires − (b) Linear density profiles extracted from absorption images. The gray curve is the linear density profile of gas confined within a quartic potential. The atomic cloud is then illuminated during $30\mu s$ by a near resonant light beam, shaped using a DMD. The resulting density profile after a time of flight of 1ms is depicted in yellow.

## 3   GHD predictions

The above experimental setup is described theoretically as follows. Under time evolution, the initial sharp boundary of the cloud gets smoother, and the time derivatives of local quantities decrease. After some time, upon coarse-graining, one expects that the gas can locally be described by stationary states. Stationary states of the Lieb-Liniger model are entirely characterized by their rapidity distribution $\rho(\theta)$. Equivalently, they can be characterized by a function $v(\theta)$ dubbed 'occupation factor' which takes values between 0 and 1, and which is related to the rapidity distribution $\rho(\theta)$ by

$$v(\theta) = \frac{\rho(\theta)}{\rho_s(\theta)}, \qquad \text{where} \qquad \rho_s(\theta) = \frac{m}{2\pi\hbar} + \int \frac{d\theta'}{2\pi}\Delta(\theta - \theta')\rho(\theta'), \tag{1}$$

and $\Delta(\Theta) = 2g/(g^2/\hbar + \hbar\Theta^2)$ is the 'scattering shift' in the Lieb-Liniger model. The functions $v$ and $\rho$ are in one-to-one correspondence and in the following we use alternately $\rho$ or $v$. [For an introduction to this formalism, we refer to the lecture notes [38] or to Section 1 of the review article [30].]

Since we assume local stationarity, the system as a whole is described by a time and position dependent rapidity distribution $\rho(x, t, \theta)$, or equivalently by the time and position-dependent occupation factor $v(x, t, \theta)$. The latter leads to simpler calculations, while the former is particularly useful to extract the linear density, which reads

$$n(x, t) = \int d\theta\, \rho(x, t, \theta). \tag{2}$$

The GHD equations [8,9] predict the time evolution of $\rho(x, t, \theta)$, or equivalently of $v(x, t, \theta)$. When written in terms of the occupation factor $v(x, t, \theta)$, the GHD equations take the form of a convective equation

$$\frac{\partial v}{\partial t} + v_{[v]}^{\text{eff}}\frac{\partial v}{\partial x} = 0, \tag{3a}$$

and a second relation that fixes the effective velocity $v_{[v]}^{\text{eff}}$ as a functional of the local rapidity distribution,

$$v_{[v]}^{\text{eff}}(\theta) = \theta - \int \Delta(\theta - \theta')\left(v_{[v]}^{\text{eff}}(\theta) - v_{[v]}^{\text{eff}}(\theta')\right)\rho(\theta')d\theta'. \tag{3b}$$

More precisely, Eq. (3a) is the 'Euler-scale' form of GHD, a diffusionless equation that is valid at the large scales. Diffusive corrections that enter in the form of a Navier-Stokes-type term [39–42], or even dispersive corrections [43], have also been studied theoretically. However, they are subleading and so far they have not been observed experimentally. We will see below that our experimental data obey the scaling collapse expected at the Euler scale (Fig. 4), so these effects seem to be negligible in our situation, at least for the analysis of the boundary profiles. This is also compatible with a recent theoretical study in the weakly interacting regime that has concluded that diffusive effects should be very small [44]. Therefore, in this paper we ignore the possibility of subleading diffusive effects (as well as all higher-order effects) in our modeling, and we stick to the Euler-scale GHD equation above.

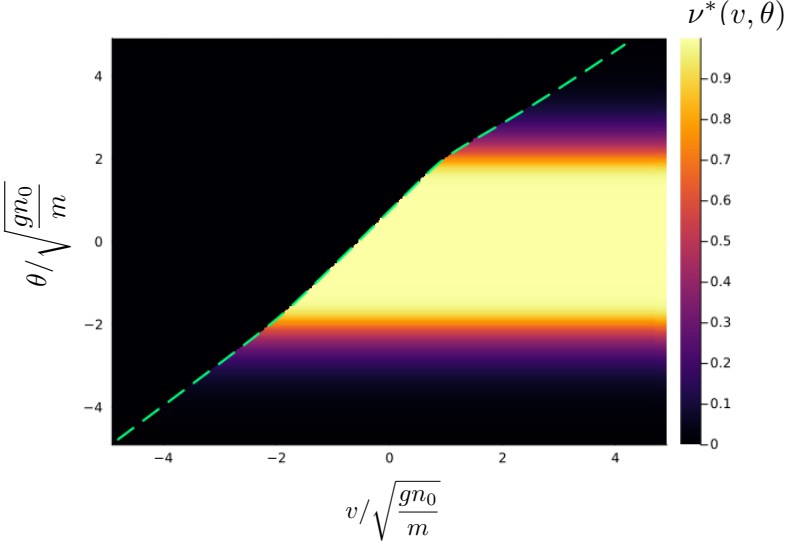

Figure 2: Occupation ratio $\nu^*(v, \theta)$ solving the equation (5) for an initial occupation ratio $\nu_0(\theta)$ in the right half-system corresponding to thermal equilibrium at temperature $T$. The dashed green line is the curve $\theta^*(v)$, *i.e.* it is the set of points $(v, \theta)$ such that $v^{\text{eff}}_{[\nu^*(v,.)]}(\theta) = v$. [Parameters: $\gamma_0 = mg/(n_0\hbar^2) = 0.005$, $k_B T\hbar^2/(mg^2) = 365$, close to the experimental parameters of the data sets below.]

For an initial bipartition whose discontinuity is located at $x = 0$, the solution of (3) is invariant along rays of constant velocity $x/t$ [8, 9]. In other words, Eq. (3) implies that, for this class of initial states, the local occupation factor distribution, and thus all local properties of the gas, depend on $x$ and $t$ only through the quantity $v = x/t$. The solution of Eq. (3) can thus be written using the occupation factor along rays $\nu^*(v, \theta)$ such that

$$\nu(x, t, \theta) = \nu^*(x/t, \theta). \tag{4}$$

For the situation considered in this paper with, initially, a vacuum state for negative $x$ and a state of occupation factor distribution $\nu_0$ one the right, the solution $\nu^*(v, \theta)$ is parameterized by an edge rapidity $\theta^*$ according to [8, 9]

$$\nu^*(v, \theta) = \begin{cases} \nu_0(\theta) & \text{if} \quad \theta < \theta^* \\ 0 & \text{if} \quad \theta > \theta^* \end{cases} \quad \text{where} \quad v^{\text{eff}}_{[\nu^*(v,.)]}(\theta^*) = v. \tag{5}$$

This equation can be solved numerically for any given initial distribution $\nu_0(\theta)$, see Fig. 2 for an example. Together with Eq. (4), it entirely describes the system at the Euler scale. Note that, to compute the linear density $n(x, t)$ in order to compare with experimental density profiles, one uses Eq. (2).

**Solution for a system initially in the ground state.** To illustrate the above formalism, let us explore its implications for the special case where the right half-system is initially in the ground state. In that case the initial occupation factor $\nu_0(\theta)$ is a Fermi sea: $\nu_0(\theta) = 1$ for $|\theta| < \Delta\theta_0$, and $\nu_0(\theta) = 0$ otherwise. The Fermi radius $\Delta\theta_0$ depends on the initial linear density $n_0$ through Eq. (2) [29].

In that case the general features of the function $\nu^*(\nu, \theta)$ that solves the GHD equation (3) are as follows ( see Fig. 3). It comprises three regions: an 'empty region' far on the left with vanishing atom density, a 'filled region' far on the right where the density is equal to the initial density $n_0$, and a 'central region' where the atom density interpolates between 0 and $n_0$. It is easy to see that the left endpoint, where the atom density vanishes, is at $\nu = -\Delta\theta_0$. The right endpoint velocity on the other hand is the sound velocity in the fluid of density $n_0$, given by $c = \nu_{[\nu_0]}^{\text{eff}}(\Delta\theta_0)$. In the central region $-\Delta\theta_0 < x/t < c$, the gas is locally in a state that is a Fermi sea shifted by a Galilean boost of velocity $V(x/t)$ for some function $V$. For arbitrary interaction strengths $g$ the density profile $n(x/t)$ cannot be computed in closed form, but it is easily computed numerically. Analytical expressions are available in the two asymptotic regimes of strong and weak interactions which correspond to $\gamma \gg 1$ and $\gamma \ll 1$ respectively.

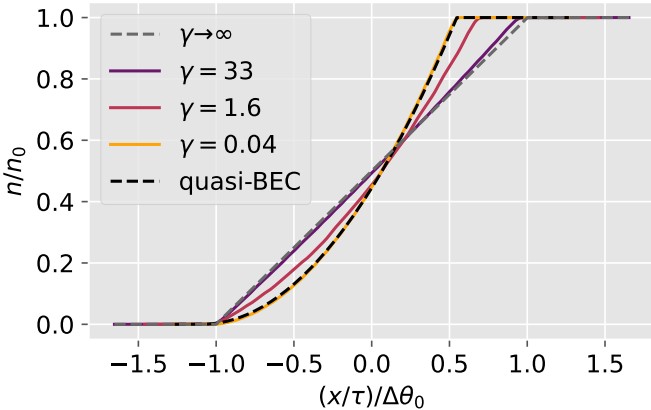

Figure 3: Boundary profile predictions from GHD for system initially in the ground state as a function of $\gamma$. The velocity is normalized to the radius of the intial Fermi sea $\Delta\theta_0$. On the negative side the point where $n$ reaches 0 is at $\Delta\theta_0$ whatever $\gamma$. On the positive side, the point where $n$ reaches $n_0$ is at the speed of sound $c$. The black, resp. grey, dashed line corresponds to the hydrodynamic prediction in the quasi-BEC regime (Eq. (7)), resp. in the hard-core regime (Eq.(6)).

In the strong repulsion regime, or hard-core regime, $\nu_{[\nu]}^{\text{eff}}(\theta) = \theta$ regardless of the occupation factor $\nu(\theta)$. Then Eq.(5) is easily solved. We can use the fact that, in this regime, a Fermi sea of radius $\Delta\theta$ corresponds to a linear density $n = m\Delta\theta/(\pi\hbar)$ to derive

$$(\gamma \gg 1) \qquad n(x,t) = \frac{n_0}{2}\left(1 + \frac{xm}{t\pi\hbar n_0}\right) \qquad \text{if} \quad -\pi\hbar n_0/m < x/t < \pi\hbar n_0/m. \qquad (6)$$

We can easily check that we recover the result expected for a gas of free fermions, as expected from the mapping of the hard-core bosons to fermions, which preserves the density [45].

In the weakly interacting regime, or quasi-BEC regime, the effective velocity at the edge of a Fermi sea of radius $\Delta\theta$ is $\Delta\theta/2$, in the frame where the Fermi sea is at rest. Then, using the fact that, in this regime, a Fermi sea of radius $\Delta\theta$ corresponds to a linear density

$n = m\Delta\theta^2/(4g)$, we obtain

$$(\gamma \ll 1) \qquad n(x,t) = n_0 \left( \frac{2}{3} + \frac{1}{3}\frac{x}{t}\sqrt{\frac{m}{gn_0}} \right)^2 \qquad \text{if} \quad -2\sqrt{gn_0/m} < x/t < \sqrt{gn_0/m}. \quad (7)$$

Here we recover the hydrodynamic predictions derived from the Gross-Pitaevskii equation [46, 47]. This is expected, since the Gross-Pitaevskii approach becomes exact in the limit of weak interactions, so it should agree with GHD, because GHD is the correct hydrodynamic equation for all repulsion strengths.

In Fig. 3, we compare the GDH solution for systems initially in the ground state with the two above asymptotic formulas. We find that the quasi-BEC regime is reached to an excellent approximation already for $\gamma = 0.04$.

## 4 Experimental data

The boundary profiles for evolution times varying between $\tau = 10$ms and $\tau = 18$ms are shown in Fig.4(a) and are represented as a function of $v = x/\tau$. The profiles overlap remarkably well, showing that the Euler scale is reached within this time interval. The longitudinal dynamics after $\tau = 18$ms cannot be probed due to the fact that our initial semi-homogeneous gas has a finite size. For shorter deformation times, experimental boundary profiles are smoother than the Euler-scale GHD predictions, which might be due to the failure of Euler scale, and/or to the fact that the cut at $t = 0$ is not infinitely sharp.

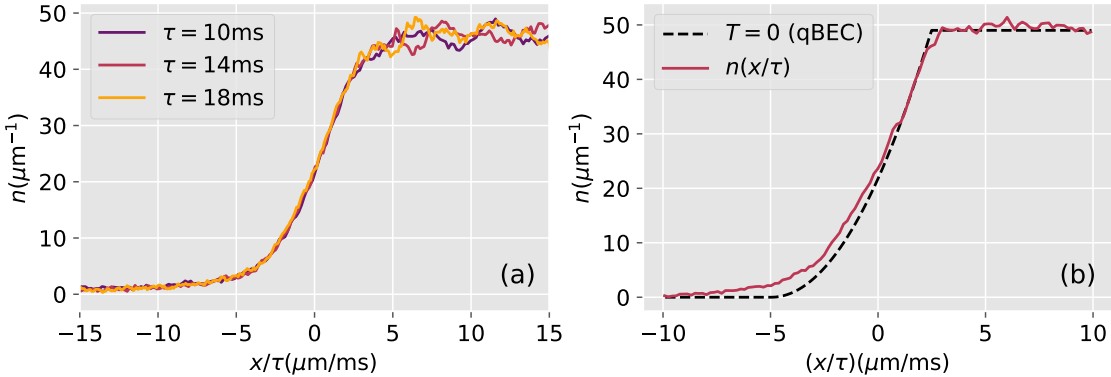

Figure 4: (a) Boundary density profiles obtained for different evolution times $\tau$ and represented as a function of $x/\tau$ — (b) Comparison of the boundary profile with the zero-temperature GHD prediction in the quasi-BEC regime Eq.(7). The latter is an excellent approximation of the zero-temperature prediction for the interaction parameter of the data which is $\gamma = 4.6 \times 10^{-3}$ (see Fig. (3)). The edge profile in figure (b) is obtained after an expansion time of 10 ms and belongs to a different data set to those shown in figure (a).

Fig.4(b) compares the measured boundary profile $n(v)$ to the GHD prediction assuming that the initial state on the right is the ground state. Here the Lieb parameter is $\gamma = 4.6 \times 10^{-3}$, and for such small values the boundary profile is indistinguishable from the quasi-condensate prediction, see Fig.3, so we actually compare the data to Eq. (7). The agreement between the ground state prediction and experimental data is rather good, especially in the high density part. The deviations from the parabola observed experimentally are due to non-zero entropy effects, which are investigated in the following section.

# 5 Retrieving the initial rapidity distribution from the boundary profile

In Section 3 we saw that, for a given initial occupation factor $v_0(\theta)$, we can compute the boundary profile $n(v)$ with the Euler-scale GHD equations. Since, in the experiment, we measure the boundary profile, it is natural to ask whether the converse operation is possible: Can we retrieve the occupation factor $v_0(\theta)$ from the boundary profile, relying on the Euler-scale GHD equations?

**Direct reconstruction.** We assume that we have a boundary profile $n(v)$ which is monotonically increasing, with $n(v) = 0$ when $v \to -\infty$, and $n(v) = n_0$ when $v \to +\infty$. A first idea is to reconstruct the function $v_0(\theta)$ incrementally, from negative values of $\theta$ to positive ones, by 'reading' the boundary profile $n(v)$ from left to right. We can start from some highly negative velocity $v_0$, such that $n(v)$ is extremely small for all $v \leq v_0$ so it can be assumed to (numerically) vanish: $n(v) = 0$ for all $v \leq v_0$. We work with discrete values of the rapidities, with a constant spacing $\delta\theta > 0$,

$$\theta_j = v_0 + j\delta\theta, \qquad j \in \mathbb{N}, \tag{8}$$

and we reconstruct the corresponding values of the occupation factor $v_j$ ($\simeq v_0(\theta_j)$) inductively. We initialize the sequence as

$$v_0 = 0. \tag{9}$$

At the $j^{\text{th}}$ step, all the occupation factors $v_0, v_1, \ldots, v_{j-1}$ are known, and we want to compute $v_j$. We fix $v_j$ by requiring that

$$n(v_j) = n_j, \tag{10}$$

where $n(v)$ is the given boundary profile, and $n_j$ and $v_j$ are numerical estimates of the particle density and of the effective velocity respectively, obtained by discretizing the various integrals that enter the definitions of Section 3:

$$n_j = \sum_{a=0}^{j} \frac{\delta\theta}{2\pi} v_a 1_{j,a}^{\text{dr}} \quad \left( \simeq \int_{-\infty}^{\theta_j} \frac{d\theta}{2\pi} v(\theta) 1^{\text{dr}}(\theta) \right)$$

$$v_j = \frac{\text{id}_{j,j}^{\text{dr}}}{1_{j,j}^{\text{dr}}} \quad \left( \simeq \frac{\text{id}^{\text{dr}}(\theta_j)}{1^{\text{dr}}(\theta_j)} \right).$$

Here $\text{id}(\theta) = \theta$ is the identity function, and the discretized dressed function $f_j^{\text{dr}}$, for a function $f$, is the solution of the linear system

$$f_{j,a}^{\text{dr}} = f(\theta_a) + \sum_{b=0}^{j} \frac{\delta\theta}{2\pi} \Delta(\theta_a - \theta_b) v_b f_{j,b}^{\text{dr}}.$$

This is the discrete analog of the definition of the dressing, which is the solution of the integral equation $f^{\text{dr}}(\theta) = f(\theta) + \int_{-\infty}^{\theta_j} \frac{d\theta'}{2\pi} \Delta(\theta - \theta') v(\theta') f^{\text{dr}}(\theta')$ [see e.g. Refs. [30, 38] for introductions to this formalism]. The value of $v_j$ that fulfills Eq.(10) can be found numerically with a root-finding algorithm; we use the bisection method.

In the limit of small spacing $\delta\theta$, this procedure is expected to converge to a continuous occupation factor $v_0(\theta)$. We have tested this method starting with the boundary profiles corresponding to some known occupation factors, and verified that it reconstructs the correct occupation factor $v_0(\theta)$ as expected.

However, when we try to apply this method to experimental boundary profiles, we face two difficulties. First, from spares and noisy experimental data points one needs to extract an increasing continuous function $n(v)$. For this, we need to fit the data with some ansatz for the boundary function. Second, we have observed that this method is highly sensitive to the details of the boundary profile $n(v)$, especially to the left tail of $n(v)$ at negative values of $v$. Since the signal-to-noise ratio in our experimental data is poor in this region, the results obtained with this technique are not trustworthy. Thus, we prefer to use an alternative method, which we present now.

**Fitting the occupation factor $v_0(\theta)$.** In order to extract the occupation factor distribution $v_0(\theta)$, we fit the experimental boundary profile with the GHD calculations based on Eqs. (4)-(5). Extracting $v_0(\theta)$ exactly would correspond to a fit with infinitely many fitting parameters, which we are not able to do. So we choose an Ansatz for $v_0(\theta)$, parameterized only by a few fitting parameters.

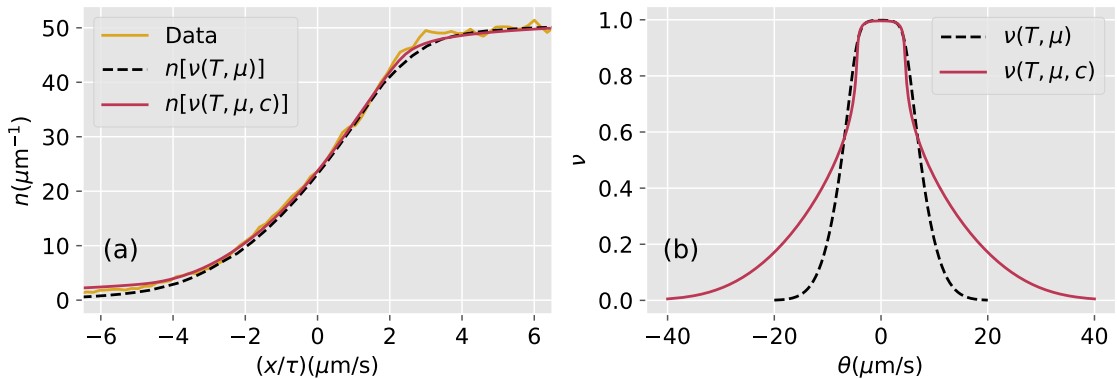

Figure 5: (a) The experimental boundary profile plotted in yellow is compared to fitted profiles using for $v_0(\theta)$ either a thermal ansatz, *i.e.* the solution of Eqs. (11) and (12), (black dashed line) or the three-parameters ansatz defined by Eqs.(11) and (13) (red line). (b) Comparison of the occupation factors obtained for both fitted occupation factor distributions.

The first ansatz that we try is the occupation factor of a Gibbs ensemble, where the fitting parameters are the temperature $T$ and the chemical potential $\mu$. This was calculated first by Yang and Yang [1], who showed that the occupation factor $v(\theta)$ is the solution of the integral equation

$$s'(v(\theta)) = \frac{\mu}{k_B T} - \frac{m\theta^2}{2k_B T} + \int d\theta' \Delta(\theta - \theta') \left[ s(v(\theta')) - v(\theta')s'(v(\theta)) \right] \qquad (11)$$

where the function $s : [0:1] \to \mathbb{R}$ is

$$s(y) = -(y \ln(y) + (1-y)\ln(1-y)), \qquad (12)$$

and $s'$ is its derivative. The integral $\int s(v(\theta))\rho_s(\theta)d\theta$ is the entropy per unit length of the occupation factor distribution $v(\theta)$ [1]. For a given $T$ and $\mu$, Eq. (11) can be solved numerically iteratively very efficiently using the fact that $s'^{-1}(\epsilon) = 1/(e^{-\epsilon/(k_B T)} + 1)$.

In Fig. 5 we compare the experimental boundary profile with the best fit obtained from this thermal equilibrium ansatz. For the data set shown here, the fitted temperature and chemical potential are $T = 282$nK and $\mu/k_B = 71.5$nK. The fit is quite good although we see some

discrepancies: the left tail of the experimental data is wider than that of the fit while on the right side, the experimental date are more sharp. Such a behavior is seen on all our data sets.

The discrepancy between the thermal fit and the experimental boundary profile may indicate the fact that the initial cloud, which is in a global stationary state in the quartic potential, is not in a thermal equilibrium state. Global stationarity is supported by the fact that the density profile of the trapped cloud is time-independent. General stationary states of the GHD equations in presence of a confining potential $V(x)$ have a local occupation factor distribution $\nu(x,\theta)$ which obeys Eq. (11) with a global 'temperature' $T$ and a local 'chemical potential' $\mu(x) = \mu_0 - V(x)$, where $\mu_0$ is the central chemical potential. However, general stationary states differ from thermal states by the choice of the 'entropy' function $s(y)$, which does not need to be given by Eq.(12), but can be an arbitrary function [48]. In the following we generalize the thermal stationary state by modifying the function $s$ as follows,

$$s(y) = -(1 + cy)(y \ln(y) + (1 - y)\ln(1 - y)), \tag{13}$$

where $c$ is a third fitting parameter (together with the 'temperature' $T$ and the 'chemical potential' $\mu$), which is vanishing for thermal states. More precisely, we fit the experimental boundary profile with the prediction for an initial occupation factor $\nu_0$ which is the one which obeys Eq. (11) with the above function $s$. A fit of our data with this ansatz gives $\mu/k_B = 74$nK, $T/k_B = 480$nK and $c = 2.92$. This fit decreases the square distance to the data by 30 % compared to the thermal fit.

## 6 Probing the local rapidity distribution

For an initial state of the gas corresponding to a smooth occupation factor $\nu(\theta)$ —for instance a thermal state—, the occupation factor $\nu^*(x/t, \theta)$ at fixed ratio $x/t$ is expected to be highly asymmetric as a function of $\theta$, according to Eq.(5). Indeed, on the right side – large $\theta$ – it has a jump discontinuity, similar to the one of the ground state occupation function, while on the left side – small $\theta$ – it is smooth. To reveal such peculiar features of the local state of the gas, we use the protocol introduced in Ref. [31] to probe the local rapidity distribution, as we explain next.

First, we let the gas expand for a time $t = 18$ ms, such that the boundary broadens and covers a large zone of $\sim 350 \mu$m, see Fig. 6(a). Then we select the slice of the gas that lies in an interval $[x_0 - \ell/2, x_0 + \ell/2]$, removing all atoms lying outside the slice with a pushing beam [31]. In Fig. 6(a) we show the density profile 1ms after the selection of the slice. The fit to a smoothened rectangular function gives $x_0 = 18 \mu$m. For calculations, the width $\ell$ will be determined using the number of selected atoms (see below). Finally, we let this slice expand in 1D for an expansion time $\tau$, and then we measure the longitudinal density $\tilde{n}(x, \tau)$. The latter reflects the total rapidity distribution of the slice $\Pi(\theta) = \int_{x_0 - \ell/2}^{x_0 + \ell/2} \rho(x, \theta)dx$, because the asymptotic behavior as $\tau \to \infty$ is $\tau \tilde{n}(x, \tau) \simeq \Pi((x - x_0)/\tau)$. The expected asymmetry of $\Pi$ is thus expected to induce an asymmetry of the density $\tilde{n}(x, \tau)$ as a function of the position $x$. We observe this asymmetry experimentally in our expansion profiles, as expected. This is shown in Fig. 6 (b) for an expansion time of $\tau = 30$ ms.

To go beyond this qualitative observation, we perform an Euler-scale GHD calculation of the expansion profile, assuming that the initial state is thermal. We extract the temperature by fitting the boundary profile before the selection of the slice, as shown in Fig. 6(a), yielding $T = 560$ nK. The chemical potential is adjusted so that the initial linear density is the linear density in the region $x > 0$ measured before the boundary broadening. Starting from the

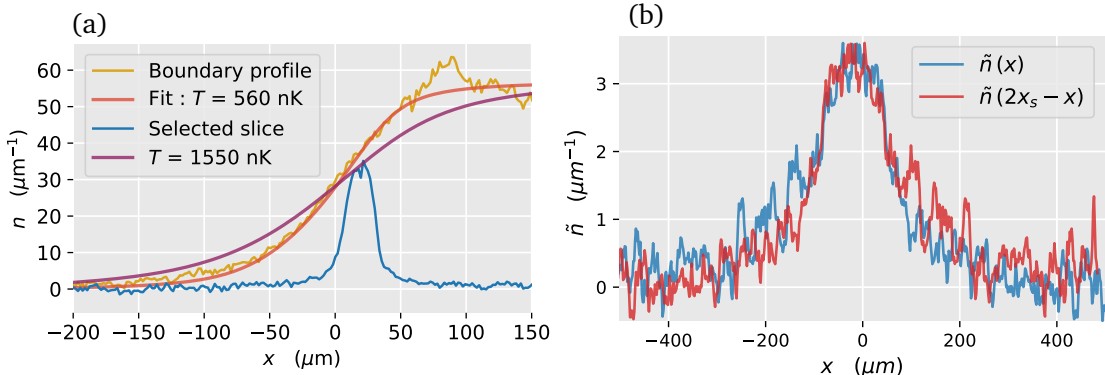

Figure 6: (a). *Boundary profile and selected slice.* The boundary profile after $18\,ms$ evolution time is shown in solid yellow line. A thermal fit yielding $T = 560\,nK$ is shown in orange. The density profile taken $1\,ms$ after the slice selection is shown in blue. The purple line is the boundary profile for the temperature deduced from a fit of the expansion profile (see text and Fig.7(b)). (b). *Asymmetry of the slice expansion profile.* The density profile after an expansion of the slice aver a time $\tau = 30$ ms is compared to its miror image. The symmetry center $x_s = -17\,\mu$m is the point that minimises the curves square distance $\delta^2 = \int dx(\tilde{n}(x) - \tilde{n}(2x_s - x))^2$.

initial sharp profile, we simulate both the boundary broadening and the slice expansion with GHD, assuming a perfect slicing, i.e. $v(x, \theta) = 0$ if $|x - x_0| > \ell/2$ and $v(x, \theta)$ is unchanged if $|x - x_0| < \ell/2$. The slice width $\ell$ is adjusted so that the calculated number of selected atoms equals the number of atoms in the experimental expansion profile, and we find $\ell = 24\,\mu$m.

The simulated expansion profile is shown in Fig. 7(a). It displays a strong asymmetry, as expected, with a sharp right edge and a vanishing density beyond a certain point on the right. The sharpness of this edge is, however, less pronounced than the one expected for the local rapidity distribution $\rho(x_0, \theta)$ at $x = x_0$, see Fig. 7(a). Two effects contribute to the broadening of the edge. First the rapidity distribution is not homogeneous inside the slice and $\Pi(\theta)$ differs from $\ell\rho(x_0, \theta)$, as seen comparing the solid brown line and dashed line in Fig.7 (a). Second, the expansion time is finite and the expansion profile is not exactly $\Pi((x - x_0)/\tau)/\tau$, as seen comparing the red and brown solid lines in Fig.7 (a).

Next, we compare the expansion profile simulated with GHD to the experimental data. As shown in Fig 7(b), the predicted profile reproduces the main features of the experimental expansion profile. Discrepancy are however as large as 25% in the central part of the profile. In an attempt to get a better agreement between data and calculations, we fitted the experimental expansion profile with the GHD calculation using the temperature of the initial state as a fitting parameter. The result, shown as magenta line in Fig.7 (b), gives a temperature $T = 1550$ nK, more than twice larger than that obtained fitting the boundary profile. The boundary profile computed for this temperature is not compatible with the experimental one, as seen in Fig.7(b).

One reason for the failure of our attempts to reproduce the density profile after the slice expansion is the presence of tails in the right edge of the experimental profile – see the density profile in Fig.7(b). Such tails are absent from the Euler-Scale GHD calculations because the occupation factor distribution inside the slice strictly vanishes above a certain rapidity. The reason for the presence of such tails is unclear. It might be due to edge effects associated to the slicing procedure, atoms at the edges of the slice being heated by the pushing beam. There is also maybe an effect of diffusion that go beyond Euler-scale GHD: the diffusive term, neglected within Euler-scale GHD, could have an impact at the beginning of the edge deformation when

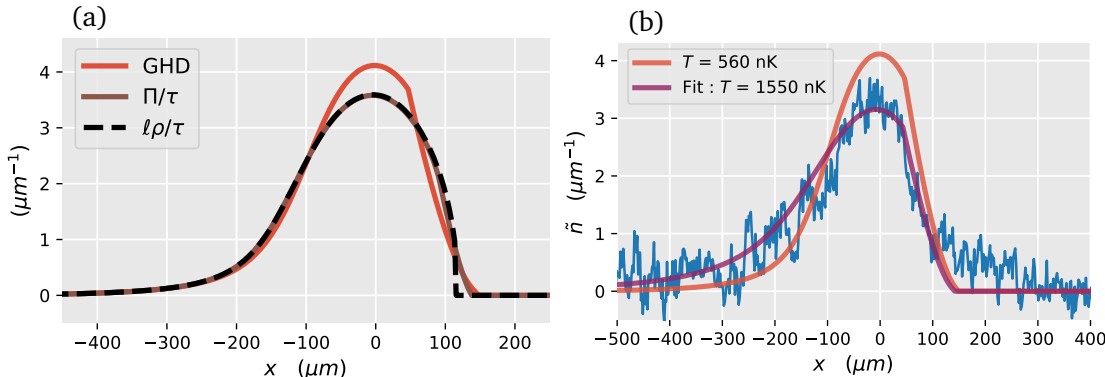

Figure 7: (a) *Density profile after slice expansion: effects of finite slice width and finite expansion time.* Orange line: GHD calculation of the density profile after an expansion of the slice for $\tau = 30$ ms, using as the initial temperature the value $T = 560$ nK obtained fitting the boundary profile. The brown line is the expected distribution if one assumes that the asymptotic large expansion time is reached, *i.e.* it shows $\Pi((x-x_0)/\tau)/\tau$, where $\Pi(\theta) = \int_{x_0-\ell/2}^{x_0+\ell/2} \rho(x,\theta)dx$ is the rapidity distribution of the selected slice. The black dashed line is $\ell\rho(x_0,(x-x_0)/\tau)/\tau$, which is the expected result in the limit of large $\tau$ and for a slice width $\ell$ negligible compared to the boundary extension. (b) *Comparison to experimental data.* Experimental data obtained after slice expansion during $\tau = 30$ ms (blue line) is compared to GHD calculations assuming an initial thermal state. The orange line, which is the same as in Fig.(a), is done for the temperature $T = 560$nK, obtained by a fit of the boundary profile. The magenta line is a fit of the experimental expansion profile yielding $T = 1550$ nK.

gradients are large.

## 7 Conclusion

We have investigated experimentally the bipartite quench protocol for a gas of bosons strongly confined transversely. We have checked that the time evolution obeys the Euler hydrodynamic scaling since the density profile is found to be a function of $x/t$ (Fig. 4). The density profile is not very far from that predicted by the generalized hydrodynamic theory for the Lieb-Liniger gas at vanishing temperature, the latter coinciding with the Gross-Pitaevski prediction for the parameters of our data. Noticeable differences are however present, due the fact the system is not in the ground state. We showed that the measurement of the boundary profile $n(v)$ could in principle permit the reconstruction of the occupation factor $\nu(\theta)$ of the initial gas, realizing a generalized thermometry method. However, in practice, we prefer to fit the observed boundary profile with the one obtained from generalized hydrodynamics using an ansatz for the occupation factor. We have found that the measured boundary profiles are not very well accounted for by a thermal occupation factor, and we have considered more general occupation factors corresponding to stationary trapped clouds, which give a better fit with our data. Finally, we present measurement of the local rapidity distribution inside the boundary. The data show the expected asymmetry of the distribution. The distribution however shows noticeable differences compared to the GHD predictions, whose origin is not elucidated.

This work calls for further experimental investigations. In the near future we plan to compare the rapidity distributions obtained with the bipartite quench protocol by fitting the

boundary profile to the rapidity distribution obtained using the slice expansion protocol [31]. Moreover, the temperatures obtained in this paper by the thermal fits are large so that the effects of populated transverse states might have an impact [49,50]. Thus it would be interesting to investigate clouds at smaller energies. The investigation of the local rapidity distribution within the border deserve further studies in order to elucidate the origin of the tails on the side that is expected to be effectively at vanishing entropy.

# Acknowledgements

We thank V. Bulchandani for discussion about general stationary states of GHD in a confining potential and A. Urilyon and J. de Nardis for discussion about the possibility to observe diffusive effects. We also thank F. Nogrette, from LCF, for work on the installation of the DMD experiment and A.-L. Coutrot, from LCF, for reparation work on the chip.

**Funding information** We thank Sophie Bouchoule, Alan Durnez and Abdelmounaim Harouri of C2N laboratory for the chip fabrication. C2N is a member of RENATECH, the French national network of large facilities for micronanotechnology. This work was supported by ANR Project QUADY-ANR-20-CE30-0017-01.

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
