# Peer review of "Experimental Investigation of a Bipartite Quench in a 1D Bose gas"

_SciPost Physics_

## Round 1 · Referee Report · Anonymous (Referee 1) · 2025-7-8

Strengths

1. The manuscript presents an innovative approach to extracting rapidity distributions in ultracold 1D quantum gases, offering a valuable alternative to existing methods and helping to bridge the gap between experiment and generalized hydrodynamics theory.
2. This work implements a clean experimental sequence for realizing bipartition, suited for the aims of the study. The paper includes an introduction to the theoretical prediction of Euler-scale Generalized Hydrodynamics (GHD), which clarifies the motivation of the experimental study and makes the work more accessible to readers not yet familiar with the topic.

Weaknesses

    1. The analysis lacks rigor: key results (e.g., fitted temperatures) show large variability without any accompanying uncertainty estimates. The implications of the quantitative results are also not addressed.
  1. The authors do not benchmark their new method against established techniques for temperature extraction (e.g. Yang-Yang thermometry, the thermometry from fluctuations, or density ripples), making it difficult to assess the method’s accuracy or sensitivity.
  2. The manuscript requires revision for completeness and precision. Some concepts, methods, and results are currently presented in a way that is either unclear or incomplete, and a number of sentences remain confusing or linguistically incorrect.

Report

The authors performed this work with a cleanly executed experiment realizing bipartition, which is a key protocol for GHD. In addition to showing that the observed ballistic dynamics follows the Eular-scale form of GHD theory, they present a method to extract the rapidity distribution of the gas. The attempt to directly connect experimental observations to the parameter used in theory is an important step for opening up future possibilities for experimental studies.

The current manuscript however has some notable shortcomings. Crucially, quantitative results are not sufficiently discussed or supported, and several claims require clarification or validation. These include, remarkably, the vastly different temperatures claimed, for which uncertainty or fitting sensitivity have not been examined. Cross-checks between the results, and discussions and interpretations of the discrepancies are lacking.

I believe after addressing the points raised and carefully revising the manuscript for completeness and clarity, the paper can properly demonstrate the values of the authors' research, and effectively convey the knowledge to the community. If the concerns are resolved satisfactorily, I would support the work for publication in SciPost Physics.

Requested changes

Questions and suggestions
1. The quantitative results claimed require error/uncertainty, and the results need to be justified and interpreted. In particular, the paper claims three temperatures acquired. The fitted values 560nK and 1550nK are very high compared to the 100nK of the prepared gas.
- The method applied for acquiring the temperature measurement/estimate of 100nK and its uncertainty need to be explicitly stated.
- The fitted temperatures and chemical potential need to have their fitting sensitivity/robustness analysed, given the fluctuations in the measured profiles.
- The meanings of the fitted temperatures need to be discussed and the discrepancies from available thermometry methods need to be addressed.
- I would like the authors to comment a bit more on the fit results, whether they are realistic or unexpected, etc.
- Justification required: Page 12, Figure 7 - Why is a thermal fit instead of a GGE (generalized Gibbs ensemble) fit applied, while the system is not globally equilibrated?
2. Cross-checking results
- Wherever possible, the temperatures presented should be checked against established thermometry methods, e.g. Yang-Yang thermometry (A. Vogler, Phys. Rev. A 88, 031603 (2013)), the thermometry from fluctuations (T. Jacqmin, Phys. Rev. Lett. 106, 230405 (2011)), or density ripples (S. Manz, Phys. Rev. A 81, 031610 (2010)).
- Page 7, Figure 4(b): Since the experiment is conducted at 100nK, it would make sense to include the finite temperature calculation for comparison.
- Page 13: “the temperatures obtained in this paper by the thermal fits are large so that the effects of populated transverse states might have an impact”. Can the authors compare quantitatively the transverse excitation energy to the tail of the rapidity distribution?
3. Concerning completeness
- Page 9, top: The authors have provided a description of the first method attempted, however this part is closed without showing explicit results. It would be helpful if the authors provide the results to demonstrate the difficulties stated.
- Page 10, Section 6: As I understand, it has not been established that local rapidity distribution can be directly probed in this work. The title “Probing the local rapidity distribution” may overstate the scope of the results. I would like to see the authors either update the results, or rephrasing the section title to avoid misleading implications. Can the authors give a succinct statement for why the comparison between rapidity distributions obtained from different approaches is not achieved already? It would be a shame that the local rapidity measurement is promised in a separate work, rather than used here to settle some of the ambiguity about the temperature/rapidity distribution.
4. Concerning clarity
- Page 5, below Figure 2: “the solution of (3) is invariant along rays of constant velocity x/t”. The stated solution is rather difficult to visualize. The authors can maybe consider describing this property of the solution in terms of rescaling the profiles on to one curve. This may also make Figure (4a) easier to understand.
- Page 10, bottom: “The chemical potential is adjusted so that the initial linear density is the linear density in the region x > 0 measured before the boundary broadening”. I did not understand this statement. Could the authors reformulate the sentence?
- Page 10 to 11: Where reference [31] is given, it would be helpful to provide a short qualitative statement why this method provides the rapidity. Because the interaction energy is released during expansion and what is observed is the asymptotic velocity? Also for explaining the slicing method, it may be helpful to show a distribution such as Figure 2, and describe what is sliced out.
- Page 12, Conclusion: “The density profile is not very far from that predicted…” “Noticeable differences…” The two consecutive statements are in apparent contradiction. Can the authors provide more careful examinations and more explicit statements in the conclusion?
5. It is stated the sharpness of the boundary is mainly limited by the imaging resolution for the pushing beam. Could this be verified by imaging the cloud after pushing? Is the cloud imaged with the same resolution, or better?
6. Page 7-9: One would think it is possible to incorporate data at different times to acquire better analysis (short times have small separation but higher SNR, longer times have lower SNR and larger separation). And the authors have specifically shown already that they have data for multiple expansion times. Have the authors considered doing so, or is there a reason that profiles at shorter times cannot be used?
7. It appears the authors’ bipartite quench operation and the resultant boundary dynamics is closely related to the work on transport “Characterising transport in a quantum gas by measuring Drude weights”, arXiv:2406.17569, by P. Schüttelkopf et al. I am interested whether the authors can make a comparison to the situation where the other side is not vacuum, and comment on what different things can be learned. in that work, the estimated equilibrium temperature appears consistent with the observed/GHD dynamics. Can the authors explain why there appears to be such discrepancy here despite the experimental platforms are similar?
8. Typing or spelling errors
Page 3 – The “fourth first coefficients” --> Please clarify
Page 3 – “could also limited”
Page 5 – occupation factor distribution v0 “one” the right --> “on” the right?
Page 9 – from “spares” and noisy experimental data points
Page 10 – the experimental “date” are “more sharp” --> “data” and “sharper”?
Page 10 – some left quotation marks have the wrong format
Page 11, Figure 6 caption – after an expansion of the slice “aver” a time --> “over”?
Page 12 - due “to” the fact that the system is not in the ground state.
Page 12 - “In the near future we plan to compare the rapidity distributions obtained with the bipartite quench protocol by fitting the boundary profile to the rapidity distribution obtained using the slice expansion protocol”. Please check this sentence again.

Recommendation

Ask for minor revision

  • validity: good
  • significance: good
  • originality: high
  • clarity: ok
  • formatting: reasonable
  • grammar: good

Author:  Jerome Dubail  on 2025-10-31  [id 5974]

(in reply to Report 1 on 2025-07-08)
Category:
remark
answer to question
correction
pointer to related literature

We thank the Referee for their careful reading and for the detailed comments and constructive feedback. Our reply is quite long so we present it in the pdf in attachment.

Attachment:

Reply_to_referee_1.pdf

---

## Round 1 · Referee Report · Anonymous (Referee 2) · 2025-7-9

Strengths

1- First experimental realization of the "bipartite quench" in the repulsive Bose gas, a notable nonequilibrium protocol originally employed to introduce the first concepts of Generalized Hydrodynamics (GHD). 2- Good agreement of experimental data and GHD 3- Introduces a new method to probe the rapidity distribution from the atomic cloud 4- Pedagogical and clearly written

Weaknesses

1.- The experiment is close to the validity of the Gross-Pitaeviskii (or classical field) approximation, therefore quantum effects are not strong.

Report

I have read with deep interest this manuscript: I am an expert in the field and it has been nice finally seeing the "bipartite quench" realized in a cold atom experiment. I think this work will be of sure interest for the broad community of researchers working on cold atoms and nonequilibrium physics, both experimentalists and theoreticians, and in particular to experts of GHD.

I can surely recommend this work for publication in Scipost Physics, and I have only some minor comments I would like the authors to consider.

1.- Since this work combines experiments and GHD, it is surprising seeing some previous interesting papers on this matter not showing up in the references. I think they should be added, in particular the first two papers on the matter: -Phys. Rev. Lett. 122, 090601 (2019) - Science373,1129-1133(2021) And the other experimental papers employing the rapidity measurement protocol (I do not mention those already included in the manuscript) -Science 367, 1461 (2020) -Science 385, 1063 (2024) -Phys. Rev. A 107, L061302 (2023) -arXiv:2505.10550 (2025)

2.- There is a choice of notation that is confusing, namely the use of "calligraphic v" -or nu, it is difficult to decipher from the pdf- for the filling function (Eq 1), for the effective velocity (Eq 3a) and for the ray x/t (above Eq 4). I think this may be confusing and it would be better introducing different symbols. A notation I have seen several times is \vartheta for the filling function and \zeta for the ray, but other choices are equally fine as long as the clarity is improved.

3.- It is the first time I see entropy modifications (see Eq 13 and discussion) to describe experimental data and it is very interesting. The authors add a "one-parameter deformation" associated to the "c" parameter in Eq. 13. The natural question is: why do the authors chose this deformation, and not another equally simple one? It would be nice if the authors could share the thoughts that inspired this choice.

4.- Below Eq 13, the authors report the value of the fitted temperature and chemical potential obtained in the presence of the deformation Eq. 13. Since this is not a thermal state any longer, I feel that calling these parameters temperature and chemical potential is deceiving and a different notation should be used, or at least a comment should be added.

5.- Bottom pg 10: is the temperature T=560nK obtained with the deformed entropy? If it is so, it should be specified.

Requested changes

See report.

Recommendation

Publish (easily meets expectations and criteria for this Journal; among top 50%)

  • validity: top
  • significance: high
  • originality: high
  • clarity: high
  • formatting: perfect
  • grammar: perfect

Author:  Jerome Dubail  on 2025-10-31  [id 5975]

(in reply to Report 2 on 2025-07-09)

We thank Referee 2 for their careful reading of our manuscript and for their comments. We are happy to see that they find this work interesting and that it deserves publication.

"Since this work combines experiments and GHD, it is surprising seeing some previous interesting papers on this matter not showing up in the references. I think they should be added, in particular the first two papers on the matter: -Phys. Rev. Lett. 122, 090601 (2019) - Science373,1129-1133(2021) And the other experimental papers employing the rapidity measurement protocol (I do not mention those already included in the manuscript) -Science 367, 1461 (2020) -Science 385, 1063 (2024) -Phys. Rev. A 107, L061302 (2023) -arXiv:2505.10550 (2025)"

Thank you for pointing out that we had omitted those references. We added them in the new version, at the beginning of Sec. 3 and Sec. 6.

" There is a choice of notation that is confusing, namely the use of "calligraphic $v$" -or $\nu$, it is difficult to decipher from the pdf- for the filling function (Eq 1), for the effective velocity (Eq 3a) and for the ray x/t (above Eq 4). I think this may be confusing and it would be better introducing different symbols. A notation I have seen several times is $\vartheta$ for the filling function and $\zeta$ for the ray, but other choices are equally fine as long as the clarity is improved."

Thank you. We have changed the notation for the `ray', or velocity, from $v=x/t$ to $\zeta = x/t$ in order to avoid possible confusion with the filling factor $\nu$. We prefer to keep the letter `$\nu$' for the filling factor, as opposed to `$\vartheta$', since we use the letter $\theta$ for the rapidities, so there would be a conflict between $\vartheta$ and $\theta$.

"It is the first time I see entropy modifications (see Eq 13 and discussion) to describe experimental data and it is very interesting. The authors add a "one-parameter deformation" associated to the "c" parameter in Eq. 13. The natural question is: why do the authors chose this deformation, and not another equally simple one? It would be nice if the authors could share the thoughts that inspired this choice."

Thank you for suggesting this. Our motivation for this Ansatz is coming from the fact that we know that the initial state is stationary in the trap, and the class of stationary states in the trap can be parameterized by such an entropy functional. This is a result of Vir Bulchandani, see Ref. [53], which we adapt here. In the revised version, we added a discussion of our motivation for this ansatz in the paragraph Eq. (13).

"Below Eq 13, the authors report the value of the fitted temperature and chemical potential obtained in the presence of the deformation Eq. 13. Since this is not a thermal state any longer, I feel that calling these parameters temperature and chemical potential is deceiving and a different notation should be used, or at least a comment should be added."

We have changed the notations to fitting parameters $a,b$ and $c$, see Eqs. (11)-(13) in the new version.

"Bottom pg 10: is the temperature T=560nK obtained with the deformed entropy? If it is so, it should be specified."

The temperature 560 nK in section 6 is obtained by the thermal fit. With the fact that we no longer use the term "temperature" for the non-thermal fit, we think it is now clear. See also the reply to Referee 1, point 1(e).

---

## Round 2 · Author Response

List of changes
(We can provide the revised version of the manuscript with all changes visible in blue upon request.)
-
In the introduction, we now mention the recent work [P. Schüttelkopf, M. Tajik, N. Bazhan, F. Cataldini, S.-C. Ji, J. Schmiedmayer and F. Møller, Characterising transport in a quantum gas by measuring drude weights, arXiv:2406.17569] where another bipartite protocol has been realized.
-
In page 3, we no longer mention the estimate of the temperature of 100nK, and the discussion of the experimental setup has been improved.
-
We have reorganized sections 3 and 4 for more clarity. Section 3 is now focused on the general theoretical prediction provided by Generalized Hydrodynamics, while Section 4 focuses on the zero-temperature (ground state) case.
-
We have improved the discussion of potential effects due to transversely excited states in Figures 5 and 6 and in the text.
-
The text in pages 10-12 has been improved so as to clarify the discussion.
-
We added an appendix to explain the details of our attempts at reconstructing the rapidity distribution from the boundary profile.

---

## Round 2 · List of Changes

(We can provide the revised version of the manuscript with all changes visible in blue upon request.)
-
In the introduction, we now mention the recent work [P. Schüttelkopf, M. Tajik, N. Bazhan, F. Cataldini, S.-C. Ji, J. Schmiedmayer and F. Møller, Characterising transport in a quantum gas by measuring drude weights, arXiv:2406.17569] where another bipartite protocol has been realized.
-
In page 3, we no longer mention the estimate of the temperature of 100nK, and the discussion of the experimental setup has been improved.
-
We have reorganized sections 3 and 4 for more clarity. Section 3 is now focused on the general theoretical prediction provided by Generalized Hydrodynamics, while Section 4 focuses on the zero-temperature (ground state) case.
-
We have improved the discussion of potential effects due to transversely excited states in Figures 5 and 6 and in the text.
-
The text in pages 10-12 has been improved so as to clarify the discussion.
-
We added an appendix to explain the details of our attempts at reconstructing the rapidity distribution from the boundary profile.

---

## Editorial Decision

editorial_decision: